# Methylation of *DROSHA* and *DICER* as a Biomarker for the Detection of Lung Cancer

**DOI:** 10.3390/cancers13236139

**Published:** 2021-12-06

**Authors:** Michał Szczyrek, Anna Grenda, Barbara Kuźnar-Kamińska, Paweł Krawczyk, Marek Sawicki, Halina Batura-Gabryel, Radosław Mlak, Aneta Szudy-Szczyrek, Tomasz Krajka, Andrzej Krajka, Janusz Milanowski

**Affiliations:** 1Department of Pneumonology, Oncology and Allergology, Medical University of Lublin, 20-950 Lublin, Poland; anna.grenda@umlub.pl (A.G.); krapa@poczta.onet.pl (P.K.); janusz.milanowski@umlub.pl (J.M.); 2Department of Pulmonology, Allergology and Respiratory Oncology, University of Medical Sciences in Poznan, 60-569 Poznan, Poland; kaminska@ump.edu.pl (B.K.-K.); pulmo@ump.edu.pl (H.B.-G.); 3Department of Thoracic Surgery, Medical University of Lublin, 20-954 Lublin, Poland; mareksawicki@umlub.pl; 4Department of Human Physiology, Medical University of Lublin, 20-080 Lublin, Poland; radoslaw.mlak@umlub.pl; 5Department of Haematooncology and Bone Marrow Transplantation, Medical University of Lublin, 20-081 Lublin, Poland; aneta.szudy-szczyrek@umlub.pl; 6Division of Mathematics, Department of Production Computerisation and Robotisation, Lublin University of Technology, 20-618 Lublin, Poland; t.krajka@pollub.pl; 7Institute of Computer Science, Maria Curie-Sklodowska University, 20-033 Lublin, Poland; andrzej.krajka@mail.umcs.pl

**Keywords:** *DICER* gene, *DROSHA* gene, DNA methylation, microRNAs, lung cancer, biomarker, detection

## Abstract

**Simple Summary:**

To identify possible biomarkers for early detection of lung cancer we assessed the methylation status of genes related to carcinogenesis, *DICER* and *DROSHA*, in lung cancer patients and healthy volunteers. The relative level of methylation of *DROSHA* was significantly lower and *DICER* significantly higher in cancer patients. The relative level of methylation of *DROSHA* was significantly higher in patients with early-stage NSCLC (IA-IIIA) and could discriminate them from healthy people with a sensitivity of 71% and specificity of 76% for the first region and with a sensitivity of 60% and specificity of 85% for the second region. Analysis of the first region of the *DICER* enabled the distinction of NSCLC patients from healthy individuals with a sensitivity of 96% and specificity of 60%. The results indicate that the assessment of *DICER* and *DROSHA* methylation status can potentially be used as a biomarker for the early detection of lung cancer.

**Abstract:**

**Background**: Lung cancer is the leading cause of cancer-related deaths. Early diagnosis may improve the prognosis. **Methods:** Using quantitative methylation-specific real-time PCR (qMSP-PCR), we assessed the methylation status of two genes (in two subsequent regions according to locations in their promoter sequences) related to carcinogenesis, *DICER* and *DROSHA*, in 101 plasma samples (obtained prior to the treatment) of lung cancer patients and 45 healthy volunteers. **Results:** The relative level of methylation of *DROSHA* was significantly lower (*p* = 0.012 for first and *p* < 0.00001 for the second region) and *DICER* significantly higher (*p* = 0.029 for the first region) in cancer patients. The relative level of methylation of *DROSHA* was significantly (*p* = 0.037) higher in patients with early-stage NSCLC (IA-IIIA) and could discriminate them from healthy people with a sensitivity of 71% and specificity of 76% (AUC = 0.696, 95% CI: 0.545–0.847, *p* = 0.011) for the first region and with a sensitivity of 60% and specificity of 85% (AUC = 0.795, 95% CI: 0.689–0.901, *p* < 0.0001) for the second region. Methylation analysis of the first region of the *DICER* enabled the distinction of NSCLC patients from healthy individuals with a sensitivity of 96% and specificity of 60% (AUC = 0.651, 95% CI: 0.517–0.785, *p* = 0.027). The limitations of the study include its small sample size, preliminary nature, being an observational type of study, and the lack of functional experiments allowing for the explanation of the biologic backgrounds of the observed associations. **Conclusion:** The obtained results indicate that the assessment of *DICER* and *DROSHA* methylation status can potentially be used as a biomarker for the early detection of lung cancer.

## 1. Introduction

Lung cancer is a global public health concern. It is the second most common malignant disease and the leading cause of cancer-related death in the world. In 2020, approximately 1.8 million deaths due to lung cancer were recorded [1]. The average five-year survival rate in lung cancer patients is <21%, and in patients with distant metastases this drops to only 2–5% [2,3]. Lung cancer is often diagnosed in the advanced stage, when clinical symptoms (e.g., cough, chest pain, haemoptysis, dyspnoea) appear and complete recovery is no longer possible.

Early detection of lung cancer improves prognosis. Screening high-risk individuals using sputum cytology or chest X-rays did not improve survival [4,5]. So far, only the low-dose computed tomography (CT) studies introduced in the US in 2011 brought a reduction in mortality due to lung cancer. A limitation of CT screening, however, is the cost, radiation exposure, and a high number of false positives [6]. Therefore, biomarkers have great potential to improve the early detection of lung cancer.

Dicer and Drosha are enzymes essential in the synthesis and maturation of microRNAs (miRNAs). miRNAs are molecules of ~20 nt in length that regulate gene expression on a posttranscriptional level. They are transcribed by RNA polymerase II (Pol II). Transcription produces transitional forms known as pri-miRNAs, which form the spatial structure of the “hairpin” [7]. Drosha is a specific double-stranded RNA endoribonuclease [8]. The primary transcript of miRNAs (pri-miRNAs) is cleaved by Drosha and a stem-loop structure of ~70 base pairs long (pre-miRNAs) is produced [8,9]. Pre-miRNAs, after transport from the cell nucleus to the cytoplasm by exportin-5, are cleaved by Dicer1—an RNase III-type enzyme. As a result, molecules without a loop connecting the 3 ‘and 5’ arms are formed, creating a double-stranded form—duplex miRNAs (miRNAs-3p and miRNAs-5p). The mature miRNAs strands are then selectively bound, along with the Argonaute AGO2 proteins, to the RNA-induced silencing complex (RISC), which plays a major role in silencing the expression of specific genes in the process of RNA interference. miRNAs, by binding to the 3’-UTR region of the target mRNA, cause translational repression or destabilization of the mRNA. In the case of full complementarity with the 3’-UTR region, the mRNA is cut. Other miRNAs have the ability to inhibit translation by attaching to the 5’-UTR region or to the RNA fragment constituting the open reading frame [10].

DNA methylation is the best-known form of epigenetic modification. It operates by adding methyl groups to the nitrogen bases of nucleotides, mainly cytosine. This process involves methyltransferase (DNMT) enzymes. DNA methylation is directly engaged in the regulation of gene expression by affecting the promoters and enhancers [11,12,13,14]. It is known that DNA methylation is involved in the process of messenger RNA biogenesis. It influences the regulation of exon assembly with pre-mRNA in the splicing process [15,16,17]. In addition, it affects alternative splicing, mediating the binding of selected proteins, including MECP2 (methyl-CpG 2 binding protein) [18].

The genes encoding microRNAs undergo similar regulation as the protein-encoding genes, which means that gene methylation or their polymorphic variants may affect the miRNAs concentration [19]. These are regulations at the DNA level; however, after transcription of the miRNAs, the quantity depends, among others, on the activity of the Drosha and Dicer enzymes. Altered miRNAs expression influences the expression of genes related to cell survival or death. As a result, disturbance of miRNAs biogenesis may induce carcinogenesis by deregulation of the cell cycle [20,21].

The aim of our study was to assess the methylation status of the *DROSHA* and *DICER* genes (in two subsequent regions according to their locations in promoter sequences) and their usefulness as biomarkers in the detection of lung cancer. There were previous studies that aimed to identify diagnostic biomarkers using methylation analysis; however, most of them focused on markers with limited diagnostic value. Identifying methylation signatures could allow to classify patients better. The secondary aim was to evaluate the correlation between the demographic and clinical factors, including survival, and the studied biomarkers.

## 2. Results

We evaluated the methylation status of four promoter regions: two for the *DROSHA* gene and two for the *DICER* gene. We observed a significant difference in the relative level of *DROSHA* gene methylation in both regions between the control group and lung cancer patients (Figure 1). There was a significantly lower relative methylation level of *DROSHA* gene in the whole group of lung cancer patients (*p* = 0.012 for the first region, Figure 1A, and *p* < 0.00001 for the second region, Figure 1B) and in NSCLC patients (*p* = 0.009 for the first region, Figure 1D, and *p* = 0.00001 for the second region, Figure 1E) compared to healthy individuals. We observed a significantly higher relative methylation level in the first region of the *DICER* promoter in lung cancer patients (*p* = 0.029, Figure 1C) and in NSCLC patients (*p* = 0.018, Figure 1F) compared to healthy people (Figure 1). However, we did not observe such differences in the methylation level of the second examined region of the *DICER* promoter in these groups of patients.

We did not find any difference in the methylation levels between the different histological types of lung cancer (*p* > 0.1).

Furthermore, we showed that the relative methylation level of the first region of the *DROSHA* gene was significantly (*p* = 0.037) higher in patients with early stage NSCLC (IA-IIIA) compared to those with locally advanced (IIIB) or metastatic (IV) NSCLC (Figure 2A). We found a significantly (*p* = 0.01) higher relative methylation level of the first examined region of the *DICER* gene in patients with larger tumours (Figure 2C). An insignificantly higher (tendency to significance, *p* = 0.053) relative methylation level of *DROSHA* (region 2) was found in patients with larger tumours (Figure 2B). An insignificantly higher methylation level of regions 1 (Figure 2D) and 2 (Figure 2E) of *DROSHA* was observed in patients with or without distant metastases, respectively (*p* = 0.081 and *p* = 0.088, respectively). 

In further analysis, we assessed the usefulness of the methylation test in distinguishing between patients in the early stages of NSCLC and healthy individuals (Figure 3). We noticed that the relative level of methylation of the first analysed region of the *DROSHA* gene could discriminate these group of patients from healthy people with 71% sensitivity and 76% specificity (AUC = 0.696, 95% Confidence Interval (CI): 0.545–0.847, *p* = 0.011). Values of these parameters in the methylation analysis of the second region of the *DROSHA* gene were as follows: 60% sensitivity and 85% specificity (AUC = 0.795, 95% CI: 0.689–0.901, *p* < 0.0001). Methylation analysis of the first region of the *DICER* gene revealed 96% sensitivity and 60% specificity (AUC = 0.651, 95% CI: 0.517–0.785, *p* = 0.027).

Based on data available in The Cancer Genome Atlas (TCGA), we performed validation of the obtained experimental results. In silico analysis revealed a significantly higher relative methylation of *DICER* in samples from patients with lung adenocarcinoma (LUAD project) compared to the control (solid normal tissue) (*p* = 0.0003; Appendix A). In turn, such an analysis shows an insignificantly higher relative methylation of *DROSHA* in samples from patients with lung adenocarcinoma compared to the control (*p* = 0.1020; Appendix A). Moreover, ROC analysis of TCGA data revealed that the relative methylation of *DICER* was characterized by 37.3% sensitivity and 77% specificity in differentiating lung cancer from the control (AUC = 0.539, 95%CI: 0.530–0.548, *p* = 0.0001; Appendix A). On the other hand, the relative methylation of *DROSHA* was characterized by 27.1% sensitivity and 82.4% specificity in the differentiation of lung cancer and the control (AUC = 0.532, 95%CI: 0.502–0.561; *p* = 0.1037; Appendix A). 

We evaluated the expression of *DROSHA* and *DICER* in 45 NSCLC patients (44.5% of the study group). In that group were 33 males (73.3%) and 12 females (26.7%). Eight patients (17.8%) were in stages IA–IIIA and 37 (82.2%) in stages IIIB–IV. Thirty-two patients (71.1%) had lymph node metastases, whereas distant metastases were observed in 34 patients (75.5%). AC and SqSC were diagnosed in 21 (46.7%) and 20 (44.4%) patients, respectively. The tumour was classified as NOS in 4 patients (8.9%).

We found no statistically significant correlation (or even a trend) between the level of methylation and the expression of *DROSHA* and *DICER* genes (*p* > 0.1). Similarly, we found no correlation between the *DROSHA* and *DICER* gene expression and demographic and clinical variables. The only exceptions were a significantly higher level of *DICER* expression in patients with early stages (IA–IIIA) compared to those in advanced stages (IIIB–IV) (*p* = 0.002, Figure 4A), and in patients without lymph node metastases (*p* = 0.045, Figure 4B) compared to those with metastatic lymph nodes.

Overall survival (OS) data were available in 61 (60.4%) patients, including 32 with AC (52.4%), 24 with SqSC (39.3%), and 4 with NOS NSCLC (6.6%). The median OS was 13.6 mth (95%CI: 8.3–19.1 mth). There were no significant differences in the median OS in the analysed subgroups, depending on the relative level of methylation of *DROSHA* and *DICER*. For *DROSHA* region 1, the median OS was insignificantly higher (14.6 mth vs. 11.3 mth) if the methylation level was above the median (HR = 0.909, 95% CI: 0.477 to 1.732, *p* = 0.771, Figure 5A). For *DOSHA* region 2, the OS was insignificantly higher (18.3 mth vs. 11.3 mth) if the methylation level was below the median (HR = 0.731, 95% CI: 0.386 to 1.384, *p* = 0.336, Figure 5B). For *DICER* region 1, the median OS was insignificantly higher (15.7 mth vs. 9.1 mth) if the median level of methylation was above the median (HR = 0.989, 95% CI: 0.522–1.871, *p* = 0.972, Figure 5C). For *DICER* region 2, the median OS was insignificantly higher (14.6 mth vs. 11.3 mth) if the median level of methylation was above the median (HR = 0.976, 95% CI: 0.5140 to 1.8519, *p* = 0.940, Figure 5D). 

## 3. Discussion

The Dicer enzyme is the major and so far the best-studied component of the miRNAs biosynthesis pathway. Altered *DICER* gene expression and activity of the Dicer enzyme have been documented in various neoplasms, such as prostate, breast, ovarian, salivary gland, central nervous system, and lung cancer, as well as sarcoma of the smooth muscle [22,23,24,25,26,27,28,29]. Interestingly, the degree of expression was different for tumours of different histological origins, suggesting both tissue and tumour-specific regulation and function of the Dicer enzyme. Studies concerning ovarian cancer patients have shown decreased expression of the *DICER* gene, while similar studies in prostate cancer, smooth muscle sarcoma, colorectal cancer, and neuroblastoma have revealed *DICER* gene overexpression and related with this a more aggressive disease course [25,26,27,28,29]. High expression of Dicer enzyme, detected by the immunohistochemistry method, and high expression of the *DICER* gene, detected by the qRT-PCR method, were unfavourable prognostic factors in patients with prostate and oesophageal cancers [25,30].

It is speculated that the high activity of the Dicer enzyme reduces the trigger of carcinogenesis in lung cancer, while after tumour development, increased expression of this enzyme triggers progression of lung cancer.

Karube et al. were the first to describe differences in *DICER* gene expression in lung cancer patients. The research was carried out in a group of 67 patients with NSCLC. The authors noted that low *DICER* expression was significantly associated with reduced survival in patients undergoing surgical resection (*p* = 0.0001). The significant prognostic value of *DICER* gene expression was confirmed in the multivariate Cox regression model, regardless of the stage of the disease (*p* = 0.001) [22].

Prodromaki et al. analysed the expression of *DROSHA*, *DICER*, and *AGO2* genes (qRT-PCR method) as well as the expression of their protein products (Western blotting, immunofluorescence, and immunohistochemistry) in five human NSCLC cell lines and in tissue samples from 83 NSCLC patients. They found that *DROSHA*, *DICER*, and *AGO2* genes were expressed in all cancerous and non-cancerous cell lines and tissue samples. The intensity of immunohistochemical staining was significantly lower in stage I tumours than in normal lung tissues. *DICER* expression was significantly higher in stage II compared to stage I and in stage III compared to stage II and I tumours. The authors suggested that high expression of *DICER* may be involved in the progression of NSCLC to advanced stages [24].

The relationship between Dicer enzyme expression status and lung cancer histology is unclear. Chiosea et al. reported expression of Dicer enzyme in different histological NSCLC subtypes using immunohistochemistry and Western blot methods. In particular, increased expression of the Dicer protein has been detected in squamous cell carcinomas compared to invasive adenocarcinomas. However, the small number of samples (n = 9) and the obtained borderline statistical significance make it impossible to draw unequivocal conclusions [23]. The above-reported differences have not been confirmed in other studies [24,31].

The results of the research conducted so far on the role of *DROSHA* expression in the pathogenesis of other neoplastic diseases are also contradictory. The downregulation of Drosha protein is associated with a poor prognosis in gallbladder cancer [32]. 

It has been suggested that disturbance in *DROSHA* gene expression in lung cancer patients may have important clinical implications.

Díaz-García et al. analysed the relationship between the expression of *DICER* and *DROSHA* genes examined by the qRT-PCR method and the presence of different histological subtypes of NSCLC as well as overall survival of patients undergoing surgery. The authors showed that low *DROSHA* expression was associated with increased median survival (154.2 versus 39.8 months, *p* = 0.016). On the other hand, high expression of this gene was associated with a decrease in the median survival in adenocarcinoma patients (*p* = 0.011), in patients with stage III (*p* = 0.038), and in patients with a low stage of the disease (*p* = 0.014). These results were confirmed in a multivariate analysis. High *DROSHA* gene expression turned out to be an independent prognostic factor of decreased survival in NSCLC patients (HR = 2.24, *p* = 0.04) [33].

All mentioned studies evaluated the expression of the *DROSHA* and *DICER* genes or their protein products in the tumour tissue or in cancer cell lines. In our work, we found that the degree of methylation of the *DICER* and *DROSHA* promoter regions evaluated in liquid biopsy (peripheral blood) differs significantly in lung cancer patients compared to the healthy population. Moreover, it was depended on the clinical advancement of the disease. We observed increased *DICER* gene methylation (first region) in patients with lung cancer and NSCLC and in large NSCLC tumours. On the other hand, we found decreased methylation within the *DROSHA* gene promoter in patients with lung cancer, with NSCLC, and with early stages (IA-IIIA) of NSCLC. The obtained results support the existing hypothesis about the functions of *Dicer* and *Drosha* in the complex mechanism of miRNA synthesis in lung cancer. Theoretically, hypermethylation of promoters causes gene silencing, while demethylation leads to their activation [13,14]. 

The importance of methylation of the *DICER* and *DROSHA* genes for the process of carcinogenesis is poorly understood so far. It seems, that disturbances in the miRNA synthesis process may play a key role in the pathogenesis of neoplastic diseases, and methylation of the gene coding the miRNA-processing enzymes may be a useful biomarker for cancer detection.

Joyce et al. analysed data from the Department of Veterans’ Affairs Normative Aging Study. The study involved 686 participants who were assessed every 3–5 years. Blood samples were analysed to measure methylation at the CpG sites throughout the genome. The methylation level of *DROSHA* was inversely associated with time to cancer development. Methylation of one CpG site in the *DROSHA* promoter region was positively associated with cancer prevalence. The authors concluded that methylation of the *DROSHA* gene may play a role in early carcinogenesis [7].

Karube et al. did not detect methylation in the promoter region of the *DICER* gene. However, they assessed only 15 samples obtained from patients with NSCLC (10 with low and 5 with high *DICER* expression) and 3 healthy lung tissue samples [22].

The limitations of our study include a small sample size, lack of sample size calculation (preliminary results), it being an observational type of study, and the lack of functional experiments allowing for the explanation of the biological backgrounds of the observed associations.

## 4. Materials and Methods

### 4.1. Study Group

We enrolled 101 subsequent lung cancer patients (aged above 18, both male and female) admitted to the clinic, to assess the methylation status of the *DROSHA* and *DICER* genes. Adenocarcinoma was diagnosed in 52 cases and squamous cell carcinoma in 34 cases. In the studied group, there were 8 cases of SCLC (small cell lung cancer), 5 cases of NSCLC NOS (non-small cell lung cancer not otherwise specified), and 2 cases of large cell carcinoma (LCC). The median age in the whole group was 65 years, with a standard deviation (SD) of 6.95 years. Tumour size data were available in 91 cases. The median tumour size was 60 mm (SD = 28.6 mm). Disease staging data were available in 93 cases. There were 4 patients in stage IA, 3 in stage IIA, 6 in stage II B, 4 in stage IIB, 10 in IIIA, 12 in IIIB, and 54 in stage IV. The characteristics of the studied group are presented in Table 1. The control group comprised 45 healthy individuals. The median age of the control group was 64 years.

### 4.2. Isolation of Circulating Free DNA from Plasma Samples

Blood samples were drawn prior to the anticancer treatment into blood collection tubes with EDTA (Ethylene Diamine Tetra Acetic acid) and immediately centrifuged at 1200× *g* for 15 min to separate the plasma. Next, the plasma was centrifuged again to remove residual cells and contamination. Then, plasma samples were transferred to the new tubes and stored in −80 °C until DNA isolation.

DNA isolation was carried out with a QIAmp Blood Midi Kit Qiagen, Düsseldorf, Hilden, Germany ) according to the manufacturer’s instructions. Samples with observed haemolysis were discarded. The minimal volume of plasma used for isolation was 4 mL.

### 4.3. Bisulfitation of DNA and Real-Time PCR

Bisulfitation of cf-DNA (circulating free DNA) samples was carried out using an EZ DNA Methylation-Direct Kit (Zymo Research, Irvine, CA, USA) according to the manufacturer’s instructions in a TPersonal thermocycler (Biometra, Germany). The quantitative methylation-specific real-time PCR (qMSP-PCR) technique was used to assess the promoters’ methylation level. Primers sequences specific for the methylated or unmethylated promoters were determined with a bisulfite-primer-seeker (Zymo Research, Irvine, CA, USA) of the investigated promoter regions. The primers for both gene promoter regions are presented in Table 2.

As a positive and negative control of the qRT-PCR, the EpiTect Control DNA and Control DNA Set (Qiagen, Düsseldorf, Hilden, Germany) was used, containing completely methylated or completely unmethylated bisulfite converted DNAs, and untreated, unmethylated genomic DNA. The reaction mixture for the real-time PCR was 20 μL and contained 12.5 μL of GoTaq^®^ qPCR Master Mix (Promega, Germany), 1 μL of forward primer, 1 μL of reverse primer (complementary to the methylated or unmethylated targeted promoter region), 3.5 μL nuclease-free water, and 2 μL of 10 ng/μL DNA after bisulfitation. The DNA concentration was measured using a BioPhotomether D30 (Eppendorf, Germany). Real-time PCR were conducted using an Eco Real-time PCR system (Illumina, Inc, San Diego, CA, USA). The reaction conditions were as follows: 10 min in 95 °C and then 40 cycles: 15 s in 95 °C, 60 s in 61 °C. The subsequent qRT-PCR reaction melt curve was included. The number of cut-off cycles taken into consideration in the analyses was in the 15 to 40 interval. Because we observed amplification in both the methylated and unmethylated set of primer product reaction, we made the following calculations: the threshold cycle in reaction with the methylation primers set minus the threshold cycle in reaction with the unmethylation primer set. In further analysis, the comparative 2^−∆Ct^ method was used to score the methylation level of the promoter region of the *DROSHA* and *DICER* gene.

### 4.4. Expression of DROSHA and DICER

We isolated mRNA from plasma using the miRNeasy Serum/Plasma Kit (Qiagen, Düsseldorf, Hilden, Germany) and reverse transcription PCR (RT-PCR) was performed using a High-Capacity cDNA Reverse Transcription Kit (Applied Biosystems, Waltham, MA, USA) according to the manufacturer’s instructions. Real-time PCR (qPCR) was performed to assess Drosha and Dicer mRNA expression on an Eco Illumina Real-Time PCR system device (Illumina Inc., San Diego, CA, USA). We used a Gene Expression Assay with Taq Man probes: Hs00203008_m1 (Applied Biosystems, Waltham, MA, USA) for Drosha and Hs00229023_m1 (Applied Biosystems, Waltham, MA, USA) for Dicer mRNA expression measurement. GAPDH was used as a housekeeping gene. The qPCR reaction contained 10 µL of TaqMan Fast Advanced Master Mix (Applied Biosystems, Waltham, MA, USA), 1 µL of TaqMan Gene Expression Assay (for *DROSHA*, *DICER* or GAPDH separate reactions), 5 µL of RNaze-free water, and 4 µL of cDNA. The reaction was performed under specific conditions: 95 °C for 20 s and 40 cycles: 95 °C for 3 s, 62 °C for 30 s. Ct values were obtained for *DROSHA*, *DICER* mRNA (cDNA), and GAPDH mRNA (cDNA). The analysis was performed using the 2^−ΔCt^ method.

### 4.5. Statistical Analysis

Due to the lack of published studies about the methylation status of the genes analysed in cell-free DNA obtained from peripheral blood of lung cancer patients, we did not perform the calculation of sample size calculation, and thus our research is of a preliminary nature.

The Mann–Whitney U test was used to compare methylation status of the *DROSHA* and *DICER* genes between lung cancer patients and the control group as well as between groups with different types of lung cancer. The test was also used to compare the relative methylation level between groups of patients with different disease stages, tumour sizes, metastasis statuses, and lymph node metastases. To evaluate the accuracy of the diagnostic test, a Receiver Operating Curve (ROC) with Area Under the Curve (AUC) was generated.

The overall survival (OS) measured in months from the diagnosis was evaluated. For the statistical analyses we used Statistica software version 13.1 and MedCalc Statistical Software version 18.11.6. Statistical significance was considered as *p* < 0.05 and *p* values ranging from 0.051 to 0.099 were considered as trends for statistical significance. The study was approved by the Ethical Committee of the Medical University of Lublin no KE-0254/219/2015.

## 5. Conclusions

Biomarkers have great potential to improve early detection of lung cancer, both alone and in combination with currently used imaging techniques. The functions of the Dicer and Drosha enzymes are inextricably linked with the synthesis of mature, endogenous miRNAs regulating mRNA stability and translation. Abnormalities in the expression of the genes encoding both enzymes have been detected in many malignancies, including lung cancer. In our study, we demonstrated that the methylation status of *DICER* and *DROSHA* significantly differs in lung cancer patients compared to the healthy population, which in the future may be used as a molecular biomarker of lung cancer.

## Figures and Tables

**Figure 1 cancers-13-06139-f001:**
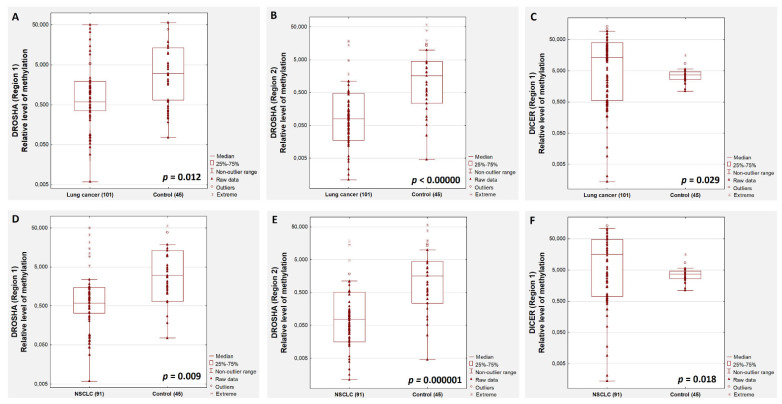
The relative methylation level of *DROSHA* and *DICER* genes in lung cancer patients (**A**–**C**) and NSCLC patients (**D**–**F**) compared with healthy individuals.

**Figure 2 cancers-13-06139-f002:**
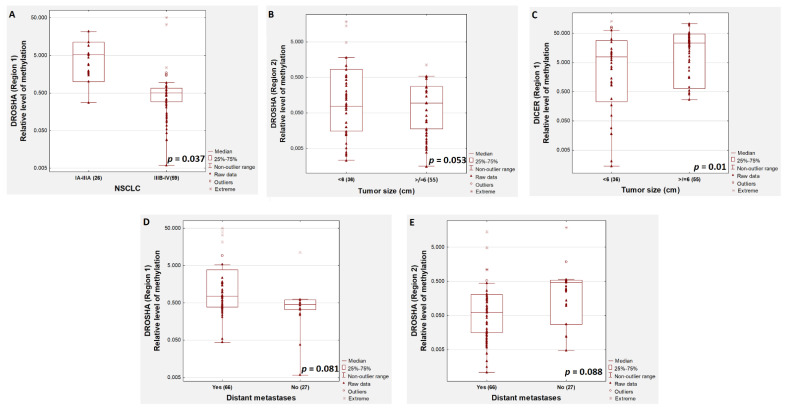
Comparison of the relative methylation level of *DROSHA* and *DICER* genes in patients with early and advanced stages of NSCLC (**A**), according to tumour size (**B**,**C**) and occurrence of distant metastases (**D**,**E**).

**Figure 3 cancers-13-06139-f003:**
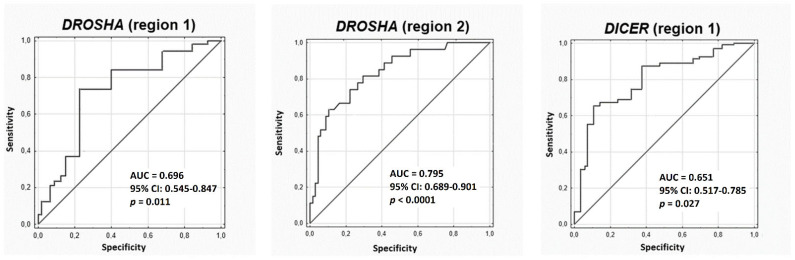
ROC curves analysis to distinguish between healthy individuals and patients with early stages of lung cancer on the basis of the relative methylation level of the *DROSHA* and *DICER* genes (description in the text).

**Figure 4 cancers-13-06139-f004:**
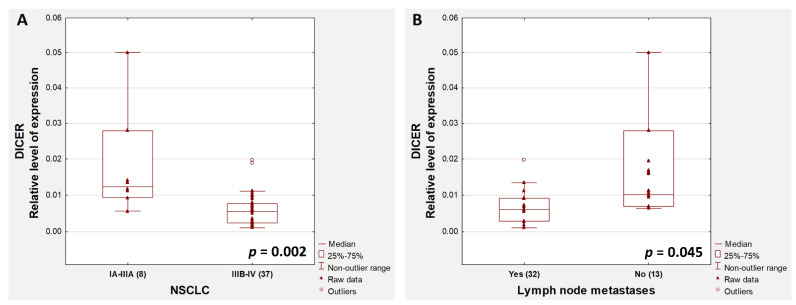
Comparison of expression of *DICER* according to NSCLC stage (**A**) and occurrence of lymph node metastases (**B**) (description it the text).

**Figure 5 cancers-13-06139-f005:**
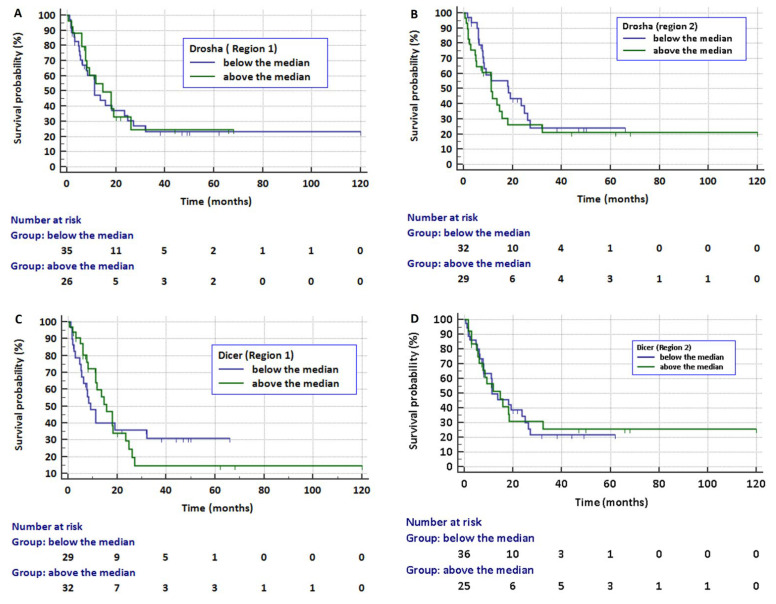
Kaplan–Meier curves representing overall survival according to the methylation level of *DROSHA* (**A**,**B**) and *DICER* (**C**,**D**) (description in the text).

**Table 1 cancers-13-06139-t001:** Characteristics of the studied group.

Characteristic	n (%)	*DROSHA1*		*DROSHA2*		*DICER1*		*DICER2*	
		Below the Median	Above the Median	Below the Median	Above the Median	Below the Median	Above the Median	Below the Median	Above the Median
**Age**									
<65	46 (45.5)	21 (45.7)	25 (54.3)	22 (47.8)	24 (52.2)	19 (41.3)	27 (58.7)	21 (45.6)	25 (54.4)
≥65	55 (54.5)	27 (49.9)	28 (50.1)	28 (50.9)	27 (49.1)	29 (52.7)	26 (47.3)	30 (54.5)	25 (45.5)
*p*-value		0.730		0.758		0.252		0.433	
χ^2^		0.119		0.095		1.311		0.615	
**Gender**									
Male	63 (62.4)	34 (54.0)	29 (46.0)	32 (50.8)	31 (49.2)	28 (44.4)	35 (55.56)	27 (42.9)	36 (57.1)
Female	38 (37.6)	14 (36.8)	24 (63.2)	18 (47.4)	20 (52.6)	20 (52.6)	18 (47.4)	24 (63.2)	14 (36.8)
*p*-value		0.095		0.680		0.425		0.048	
χ^2^		2.788		0.170		0.637		3.908	
**Histopathology**									
Adenocarcinoma (AC)	52 (51.5)	21 (40.4)	31 (59.6)	29 (55.8)	23 (44.2)	26 (50.0)	26 (50.0)	26 (50.0)	26 (50.0)
Squamous cell carcinoma (SqSC)	34 (33.7)	20 (58.8)	14 (41.2)	16 (47.1)	18 (52.9)	14 (41.2)	20 (58.8)	19 (55.9)	15 (44.1)
NSCLC NOS	5 (4.9)	3 (60)	2 (40)	3 (60)	2 (40)	2 (40)	3 (60)	3 (60)	2 (40)
SCLC	8 (7.9)	3 (37.5)	5 (62.5)	1 (12.5)	7 (87.5)	6 (75)	2 (25)	3 (37.5)	5 (62.5)
LCC	2 (2.0)	1 (50.0)	1 (50.0)	1 (50.0)	1 (50.0)	0 (0)	2 (100)	0 (0)	2 (100)
*p*-value		0.487		0.240		0.285		0.531	
χ^2^		3.443		5.501		5.024		3.161	
**Tumour size**									
<60 mm	36 (39.6)	22 (61.1)	14 (38.9)	16 (61.5)	16 (44.4)	20 (55.6)	16 (44.4)	16 (44.4)	20 (55.6)
≥60 mm	55 (60.4)	24 (43.6)	31 (56.4)	28 (50.9)	27 (49.1)	17 (30.9)	38 (69.1)	32 (58.2)	23 (41.8)
*p*-value		0.103		0.933		0.019		0.199	
χ^2^		2.658		0.007		5.478		1.647	
**Stage**									
IA-IIIA	27 (29.0)	4 (14.8)	23 (85.2)	18 (66.7)	9 (33.3)	19 (70.4)	8 (29.6)	10 (37.0)	17 (63.0)
IIIB-IV	66 (71.0)	38 (57.6)	28 (42.4)	26 (39.4)	40 (60.6)	24 (36.4)	42 (63.6)	39 (59.1)	27 (40.9)
*p*-value		0.00005		0.017		0.003		0.053	
χ^2^		16.405		5.717		8.914		3.739	
**Lymph nodes metastases**									
No	17 (31.5)	6 (35.3)	11 (64.7)	11 (64.7)	6 (35.3)	12 (70.6)	5 (29.4)	4 (23.5)	13 (76.5)
Yes	37 (68.5)	11 (29.7)	26 (70.3)	18 (48.6)	19 (51.4)	24 (64.9)	13 (35.1)	17 (45.9)	20 (54.1)
*p*-value		0.683		0.359		0.678		0.117	
χ^2^		0.167		0.841		0.172		2.463	
**Distant metastases**									
No	27 (29.0)	19 (70.4)	8 (29.6)	8 (29.6)	19 (70.4)	17 (63.0)	10 (37)	12 (44.4)	15 (55.6)
Yes	66 (71.0)	25 (37.9)	41 (62.1)	37 (56.1)	29 (43.9)	29 (43.9)	37 (56.1)	36 (54.5)	30 (45.5)
*p*-value		0.004		0.021		0.096		0.376	
χ^2^		8.115		5.360		2.774		0.783	

**Table 2 cancers-13-06139-t002:** Sequence of primers used in qMSP-PCR.

	*DICER*	*DROSHA*
First region	MF1:	MF1:
	AAGGTTTAGTTTAGGCGTTGGGTCGTAAAC	AGAAGATTCGGGAAAGTCGGCGTTT
	MR1:	MR1:
	GCTTAAAAAATCCCACTAACTCCCGCA	CCACCGCAAAACCTTATACGCGATAA
	UF1:	UF1:
	AGGTTTAGTTTAGGTGTTGGGTTGTAAATGT	AGTTGGAGAAGATTTGGGAAAGTTGGTGTTT
	UR1:	UR1:
	ACCACTTAAAAAATCCCACTAACTCCCACA	CAACCCACCACAAAACCTTATACACAATAA
Second region	MF5:	MF2:
	GGGTTAATTAGAGTGTTTGGGATTTTAATTC	AAGTGGAGTAGTTTTAAGGAATGGTC
	MR5:	MR2:
	AAATTTACTATCAATAAATTTTTTCCACCCG	AAAATCTAACTCCATATCCTCCTCG
	UF5:	UF2:
	GGGTTAATTAGAGTGTTTGGGATTTTAATTT	AGTGGAGTAGTTTTAAGGAATGGTT
	UR5:	UR2:
	AATTTACTATCAATAAATTTTTTCCACCCA	AAAATCTAACTCCATATCCTCCTCA

## Data Availability

The data are contained within the article.

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
