# Peer review of "Methylation of DROSHA and DICER as a Biomarker for the Detection of Lung Cancer"

_cancers, 2021, doi:10.3390/cancers13236139_

Round 1
Reviewer 1 Report
The article presents very preliminary results on potential associations between genes and lung cancer. These results are quite obesrvational without any mechanistic insights. In addition, those associations are not novel, since genes have already been investigated in cancer. For these reasons, I cannot recommend to consider this study for publication.
Author Response
Reviewer 1.
The article presents very preliminary results on potential associations between genes and lung cancer. These results are quite observational without any mechanistic insights. In addition, those associations are not novel, since genes have already been investigated in cancer
Response: We would like to thank the Reviewer for the time spent reviewing our manuscript. As we are all active clinicians, our study was focused on the potential real-world implications of the investigated matter. While Drosha and Dicer both on mRNA and protein level have already been studied as potential markers in lung cancer, the methylation status of their genes (DROSHA and DICER) has not been investigated in this context so far. As noted by the Reviewer, our results are preliminary and we plan to continue the research in the future.
Reviewer 2 Report
The manuscript by Szczyrek and colleagues discusses the differences in the methylation status of DROSHA and DICER as early diagnostic biomarkers for lung cancer. Using quantitative methylation-specific PCR, they assessed the methylation status of both genes in blood samples from 101 lung cancer patients and 45 healthy donors. The paper has an interest for broad scientific auditorium and provides interesting information for lung cancer. Overall, I consider the manuscript deals with an issue of interest and fits the scope of the journal, although further analyses are required to validate their results, since they draw a conclusion from limited evidence. Some revisions that should be taken into consideration to improve the presentation of this manuscript are:
- Including data, maybe a table, on the differences in the methylation status of DROSHA and DICER according to age, gender, histopathology, etc.
- Giving more data about sample collection and patients eligibility criteria. Were samples taken prior to treatment? The abstract should refer to the fact that blood samples were used.
- Using scattered boxplots instead of boxplots.
- Validating the results in an independent cohort of patients. In silico cohorts are available if more patients cannot be recruited.
- Many studies have tried to identify diagnostic biomarkers using methylation analysis. Nevertheless, most of them are focused on single markers with limited diagnostic value. Finding methylation signatures could lead to the discovery of molecular subclasses and classify patients better (10.1158/0008-5472.CAN-06-1191, 10.1038/ng1167).
- A statement on the limitations of the study should be included in the abstract and at the end of the discussion.
Reviewer 3 Report
In this study, Szczyrek et.al. have explored the biomarker potential of DROSHA and DICER methylation. This is an interesting study on lung cancer, but it needs major improvements. Additionally, the manuscript should be thoroughly re-evaluated for the English language and correct usage of sentence structures.
Here are the main comments followed by minor comments:
1. First, the mRNA expression should be quantified to measure biologically effective functions of DICER and DROSHA. Only methylation quantification, although interesting makes it difficult to extract clinically relevant information.
2. Second, adenocarcinoma and squamous carcinoma differ at biological and clinical levels and follow a different evolutionary trajectory. A separate analysis of each cancer should be included and discussed in a separate figure in the manuscript. (Although authors have tried to show differences, for example in figure 1, they have included NSCLC and Lung cancer separately but there is a lack of squamous comparison). Similarly, all the analyzes, including ROC, should be done for each biologically/clinically relevant subset.
3. Third, validation should be addressed. Authors can use TCGA or GEO datasets to see if they explore levels or methylation levels in independent datasets. (For example comparison of TCGA datasets reveals differential expression of DICER in normal vs tumor (TCGA-Lung adenocarcinoma dataset).
4. Fourth, figures should be improved. As there are different subsets of data analyzed in each figure, it becomes important to show the total number of samples in each sub-dataset. For example, the number of patients is different in the Lung cancer subset (Figure 1 A) compared to the NSCLC subset (Figure 1 D). To overcome this disparity, authors can redraw their figures with BOXPLOTS with individual observation on top of boxes as dots. In another approach, authors can simply mention the number of patients in each subset on the x-axis.
5. The authors should mention if they had utilized any calculations/formulas to measure the sample size for this study. If not, the authors should mention it in the study design and if there are shortcomings, they should be mentioned in the shortcomings section of the study.
a. The author should expand ‘first region’ and ‘second region’ in the abstract.
b. In the abstract, AUC should be accompanied by 95% C.I.
c. Authors should expand the selection of regions in the methodology. How were these regions selected? Why were the regions limited to 2?
d. Line 48-50, the sentence should be restructured. “dose computed tomography (CT) studies introduced in 2011 raise…”
e. In Figures 1 and 2: The figures can be improved to present any outliers. Also, the figures should be accompanied by the median/min/max description in the figure legend.
f. ROC figures should be accompanied by AUC, 95% C.I., and p-value. Further, all the ROCs should be presented (for example DICER in region 2). In case authors don’t want to include all the figures in the main manuscript, they can add them in supplementary figures.
g. All the results for both regions should be presented/discussed/compared.
h. In line 104, “Furthermore, we showed that methylation relative level of the first region of DROSHA..” The sentence should be restructured.
i. The font size should be uniform throughout the manuscript.
j. Authors should present survival information, Kaplan Meier plot/Hazard ratio (OS or PFS) of their dataset. It would help in identifying the prognostic value of the methylation or expression (if mRNA expression is quantified).
k. The manuscript should be thoroughly checked for English grammar, punctuation, and sentences.
Round 2
Reviewer 1 Report
No further comments.
Reviewer 2 Report
Accept in present form
Reviewer 3 Report
The Authors have addressed all the questions.